# Use of Goal Attainment Scaling to Measure Educational and Rehabilitation Improvements in Children with Multiple Disabilities

**DOI:** 10.3390/bs13080625

**Published:** 2023-07-27

**Authors:** Kimberly Kascak, Everette Keller, Cindy Dodds

**Affiliations:** 1Office of Interprofessional Initiatives, Medical University of South Carolina, Charleston, SC 29425, USA; 2Department of Public Health Sciences, Medical University of South Carolina, Charleston, SC 29425, USA; kellerev@musc.edu; 3College of Health Professions, Medical University of South Carolina, Charleston, SC 29425, USA; doddscb@musc.edu

**Keywords:** children with multiple disabilities, goal attainment scaling, quality of life, individual education program

## Abstract

With a focus on children with multiple disabilities (CMD), the purpose of this quality improvement project was to elevate educational measurement and practices involving CMD. Using the goal attainment scaling (GAS) methodology, this project was conducted within a public charter school, Pattison’s Academy for Comprehensive Education (PACE), focusing on 31 CMD and measuring student improvement and program effectiveness. For 2010–2011 and 2011–2012, improvements were demonstrated for the majority of CMD by meeting or exceeding their goals. Goal attainment scaling was able to capture improvement in educational and rehabilitation goals in the majority of CMD. Goal attainment scaling can provide an indication of a program’s effectiveness. The use of GAS in CMD has potential to maximize participation across the school setting where all children in the United States commonly develop and learn skills as well as find meaning.

## 1. Introduction

The United Nations Convention on the Rights of Children states that “a child with mental or physical disabilities is entitled to enjoy a full and decent life, in conditions that ensure dignity, promote self-reliance and facilitate the child’s active participation in the community [1]”. Solutions to ensure that the rights of children with disabilities are upheld are multi-contextual, complicated, and often involve struggles between health care and rehabilitation as well as between health care and public-school systems that provide special education and related services (e.g., physical therapy (PT) and occupational therapy (OT)) in the least restrictive environment to ensure a free and appropriate education. Across childhood, school environments provide an essential participatory setting and context in which children develop and learn motor, self-care, social–emotional, and cognitive skills. This is not only true for typical children, but also and possibly more importantly, for children with disabilities, including those with multiple disabilities.

In the United States, the 7.2 million children with disabilities account for 15% of children enrolled in public school, and children with multiple disabilities (CMD) account for 2% of that total [2]. Descriptions and definitions for this 2% of children vary across educational and rehabilitation institutions. Through the Individuals with Disabilities Education Act (IDEA), the Department of Education defines CMD as those with concomitant impairments (such as intellectual disability–blindness or intellectual disability–orthopedic impairment) that in combination cause such severe educational needs that they cannot be accommodated in special education programs solely for one of the impairments [3]. In the field of healthcare, children with medical complexity (CMC) are most similar to CMD. Berry and colleagues [4] defined CMC as a subset of children with special health care needs who have primary diagnoses that may be acquired or congenital. They also demonstrate multisystem impairments, resulting in considerable functional activity limitations (e.g., walking, dressing, talking) and participatory restrictions (e.g., school, community); experiencing frequent and extensive hospitalizations; and requiring multiple primary care, subspecialty care, and multi-disciplinary rehabilitation visits. Poly-pharmacology, medical equipment, and care coordination are commonplace. Because this project involves children within a school “setting”, the term “children with multiple disabilities” (CMD) will be used in this paper.

The Individuals with Disabilities Education Act [3] is a federal law that ensures that children with disabilities (0–21 years) receive a free, appropriate public education in the least restrictive environment. To ensure the delivery of this education, each child served by special education has an Individualized Educational Program (IEP) that is a binding legal document. It is made up of present levels of performance based on evidence-based educational and rehabilitation assessments, special education services, related services (PT, OT, transportation, etc.), supplementary services, assistive technology (AT), and measurable annual goals. It is important to note that one single assessment cannot determine a child’s educational placement and tests cannot be discriminatory (e.g., racial, cultural, gender). The facilitation of parental participation is strongly encouraged across the IEP process.

Children with multiple disabilities have multiple bodily and environment barriers, which can challenge comprehensive assessment, objective measurement, and overall functional achievement in life. To properly evaluate the present levels of CMD, the use of multiple assessments is necessary because of the heterogeneity within the population and individual nature of each CMD’s ability. One conceptual framework that has been used to describe, measure, and guide assessment and treatment in individuals with disabilities, including CMD, is the International Classification of Functioning, Disability and Health (ICF) [5,6]. The ICF is a biosocial framework for communicating and describing health and disability. It is an interactive framework made up of the domains in body structure and function, activity, participation, and environmental and personal factors [5]. Best practice supports the use of evidence-based measurement tools to capture outcomes across that ICF. For CMD specifically, measurement across the ICF in combination with the assessment of their unique characteristics allows for ideal individualization. Because CMD have multiple system impairments, activity limitations, and participatory restrictions, multiple measurement tools are necessary to document individual abilities and goal areas while the ICF framework provides an excellent model for organizing and prioritizing goal areas and subsequent treatment strategies.

Many of the commonly used assessments target appropriate skills and functional activities for children with disabilities; however, they are not sensitive enough for the subtle and slower changes that may occur in CMD [7,8,9,10]. In addition, the majorities of these measures were not specifically developed with CMD in mind. As such, documenting change or responsiveness for change is often negligible for this population, which means improvement is either not objectively captured or there is a no perception of educational or rehabilitative improvement [9]. One measure that has been used to document change in children with disabilities within education and rehabilitation settings including school settings is goal attainment scaling (GAS) [11,12,13].

Goal attainment scaling provides an individualized and patient- and family-informed method toward achieving a desired outcome and its approach to individualization is well suited for inclusion within an IEP. Steenbeek and colleagues [14] compared positive findings from GASs, the Pediatric Evaluation of Disability Inventory (PEDI), and the Gross Motor Function Measure (GMFM)-66 of children with cerebral palsy (CP) across all Gross Motor Function Classification System levels and determined that 20% of items found within the GASs were not captured within the PEDI and GMFM-66. This suggests that GASs may better measure unique aspects of the patient and family experiences as compared with fixed-item patient-reported measures like the PEDI or GMFM [14,15,16]. In a systematic review including 52 pediatric rehabilitation studies, implications are that GAS can be effectively used across a variety of diagnoses and interventions and is reflective of meaningful change. REF Three of these studies were conducted in school environments, and eight studies involved children with CP at Gross Motor Function Classification System levels of IV and V (i.e., 111 out of 444 children), which is commonly reflective of CMD. None of the studies specifically addressed CMD or CMC. The aim of this project was to describe assessment to guide GAS development as well as the implementation and measurement/analyses of GAS integrated into the IEP of 31 CMD in a school setting.

## 2. Methods

### 2.1. Protections for Human Participants

Conducted within a public charter school for CMD, this project was considered a quality improvement with the goal of elevating educational measurement and practices involving CMD. Parents and legal guardians of participating children did consent to participation in the school that provided individualized assessment, goal creation and measurement, and associated interventional strategies. The Board of Directors approved the school’s data collection processes and annually reviewed findings from annual outcome data collection. It should be noted that 50% of the board members were employees of an academic medical center and other healthcare facilities, which supported their clear understanding and oversight concerning the consent process.

### 2.2. Project Site and Participants

Pattison’s Academy for Comprehensive Education (PACE) is a public charter school with a mission to improve the quality of life (QOL) for children with multiple disabilities by providing the high-quality integration of education and rehabilitation. It was considered a separate school by the Department of Education that served only CMD. The school was conceptualized and implemented based on the market testing of the non-profit Pattison’s Academy, which hosted 5-week summer camps for CMD. One underlying principle of the summer camp was to actively engage the sensory and motor abilities of children to encourage optimal participation across the camp day with the goal of each child being physically engaged for 180 min across the 6-h day [17]. This 180-min expectation was translated to PACE upon its opening and often was reflected in goals.

Pattison’s Academy for Comprehensive Education was made up of five classrooms with 6 to 7 CMD students. In total, 5 special education teachers, 7 educational assistants, 2 physical therapists, 2 occupational therapists, 1 speech and language pathologist, and 1 school administrator made up the faculty. The indoor and outdoor environment was fully accessible and a wide variety of assistive and adaptive equipment was available. The children attended school eight hours a day (8 a.m. to 3 p.m.) from mid-August to mid-June.

Thirty-one CMD between the ages of 4 and 16 years attended PACE in Charleston, South Carolina. Thirteen children were African American, seventeen were White, and one identified as Other. All children were non-Hispanic. Fifteen were male and sixteen were female. See Table 1 for additional demographic details.

## 3. Outcomes

Goal attainment scaling was the primary outcome used by PACE to measure student improvement and program effectiveness. Goal attainment scaling is an objective goal writing approach that allows for individualization and should involve child and family input. The standardized process is well suited for measuring educational and rehabilitation capacity and performance across the ICF [5]. Goals using GAS methodology should also follow “SMART” criteria: specific (i.e., “who”, “what”, “when”, “why”, and “where”), measurable, attainable, relevant, and realistic within a temporal boundary [18,19]. Scalings are found to be more reliable and valid when developed in collaboration with families and educational and rehabilitation providers [11,12,19].

The normal distribution of a bell-shaped curve serves as the statistical foundation for GAS. Based on family interviewing and evidence-based educational and rehabilitation assessments, levels of performance that the child is expected to achieve are assigned values equally spaced across the five levels between −2 and 2 (i.e., −2, −1, 0, 1, 2). Across the expected levels of performance, only one specific variable, such as repetitions, duration, frequency, distance, assistance level, and steps across a task (task analysis) should be measured. The −2 and +2 values correspond to 2 standard deviations below and above the expected mean, respectively. Although, several acceptable options are available for defining GAS scores [11,12,20]; this project defined −2 as the “current level of performance”, which is the most common GAS methodology used. The −1, 0, 1, and 2 values are defined as progress toward the expected level of performance, expected level of performance, greater than the expected level of performance, and much greater than the expected level of performance. Goals should be designed to achieve the expected level of performance.

Scoring requires assigning a raw score between −2 and 2 to every individual GAS. For a group of GASs, raw scores are summed. Scaled and T-scores are derived from the GAS raw scores using the statistical formula or tables entitled *Conversion Key for Follow-Up Guides Having One Scored Scale* available in *Goal Attainment Scaling: Applications, Theory, and Measurement* [11] (p. 274) or the GOALed app [21]. The T-scores are the standardized statistic that allows within or between subject comparisons. T-scores are reflective of small sample sizes (n < 30) with unknown population standard deviations, which points to individuality [22]. The interpretation of T-scores can reflect an individual’s single goal, an individual’s groups of goals, a group’s group of goals, or more broadly, an entire program’s goals. Interpretations from collective groups and program goals are reflective of a program’s effectiveness.

## 4. Procedures

As a public charter school serving children within the domain of special education, PACE required annual IEPs with measurable annual goals. Because GAS was a primary outcome of the school, goals formatted into the GASs were integrated into annual IEPs. To guide and support the development of IEP goals using the GAS methodology, four steps were carried out.

### 4.1. Step 1: Evidence-Based Assessment

#### 4.1.1. Required Assessments: These Assessments Were Administered for All Enrolled Students

##### Developmental Assessment for Individuals with Severe Disabilities

The Developmental Assessment for Individuals with Severe Disabilities (DASH) 2 [23] was the required annual educational and rehabilitation assessment that was administered to every student by the educational and rehabilitation faculty members. The DASH is a criterion-referenced assessment that measures skills across five scales (i.e., academic, sensory–motor, activities of daily living, language, and social–emotional) in children and adults who have severe physical and sensory disabilities.

##### Caregiver Priorities and Child Health Index of Life with Disabilities

With a mission to improve QOL for CMD, annually measuring QOL was important, so parents completed the Caregiver Priorities and Child Health Index of Life with Disabilities (CPCHILD) [24]. The CPCHILD has demonstrated validity and measures health-related QOL in non-ambulant children with severe developmental disabilities. It is composed of 37 questions across six domains: Personal Care/Activities of Daily Living; Positioning, Transferring and Mobility; Comfort and Emotions; Communication and Social Interaction; Health; and Overall Quality of Life. Scores are on a 5- or 6-point Likert scale with levels of assistance and intensity also documented. A higher score on the CPCHILD indicates a more enhanced HRQL.

##### Faces Pain Scale—Revised

Although 180-min of active participation for CMD has been found to be safe [17,25,26] and as such was implemented each school day, it was important to monitor pain severity and discomfort in the event that this intensity was problematic for some CMD. To measure pain severity, the validated 10-point metric Faces Pain Scale—Revised was used and reported by parents [27]. It displays gender-neutral faces in various degrees of pain. The “0” space on the scale displays a face with a neutral expression while the “most pain possible” face is described as the “10” space [28,29].

##### Pediatric Evaluation of Disability Inventory

The Pediatric Evaluation of Disability Inventory (PEDI) [30] is a valid measurement tool, which considers functional skills and caregiver assistance across the three domains of self-care, mobility, and social function. Self-care, mobility, and social function domains for both the functional skills and caregiver assistance sections were administered and scored by therapists (i.e., PT, OT, SLP) based on observations in the school setting. The use of this measure allowed for a more in-depth examination of the child’s needs for caregiver assistance and overall functional abilities, which informed goals [31,32].

#### 4.1.2. Commentary Assessments: These Assessments Were Completed When Appropriate for Individual Children

In addition to the required assessments, complementary assessments were administered according to each child’s individual characteristics and needs of the educational and rehabilitation team. Because many of the CMD presented with cortical visual impairment (CVI) and had assistive technology needs, the Cortical Visual Impairment Range Assessment and Matching Assistive Technology to Child-Augmentative Communication Evaluations Simplified Assessment (MATCH-ACES) was also administered as needed. Findings from these assessments expanded descriptive data available on each child, which better guided goal development and intervention delivery.

##### Cortical Visual Impairment Range Assessment

The Cortical Visual Impairment Range Assessment (CVIRA) [33] is a reliable functional vision assessment designed to investigate the visual characteristics associated with CVI. It measures the degree of CVI severity on a 0–10 continuum where 0 represents no visual function and 10 represents near typical visual function [34,35].

##### Matching Assistive Technology to Child-Augmentative Communication Evaluations Simplified Assessment

The Matching Assistive Technology to Child-Augmentative Communication Evaluations Simplified Assessment (MATCH-ACES) [36] is an evidence-based comprehensive evaluative process that is child-centered and is used to identify the need for assistive technology in the educational setting. The administration of the MATCH-ACES was found to be an effective process for appropriately matching students with assistive technology (AT) in order to meet educational goals [36]. To ensure data obtained from each child were accurate and best reflected their abilities, no more than two assessments occurred in a day and the completion of all assessments occurred over 2 weeks at a minimum.

### 4.2. Step 2: Family Interview

As outcome measures were being administered, the educational and rehabilitation team also interviewed parents/legal guardians to identify and discuss skills and goals that were important to the child and family. These interviews were conducted at the school or in children’s homes. Again, because PACE’s mission was to impact QOL, a school-specific QOL questionnaire was developed that was derived from a QOL model created by Patrick and Chiang [37,38]. See Appendix A for this questionnaire. This questionnaire served as a script or guide for enhancing discussion between the family and school faculty. It was not required that each question had to be asked and answered, but it was essential that the information gathered was thorough and meaningful to families. This process ensured that created goals were collectively relevant for the home, community, and school environments. For example, one child’s grandmother was interested in her grandchild learning to manage her drooling and the school team agreed this was a great idea. As such, a goal attainment scaling was created to address the skill. Similarly, the grandmother wanted the child to learn to operate a remote control for the home television channel selection and although this was not an appropriate school goal, we were able to incorporate switch access into a communication device and school socialization goal.

### 4.3. Step 3: ICF and GAS Development

Following the assessment and family interview phases, all findings were organized across the ICF framework. See Appendix B for an example involving a CMD. This summative document was instrumental in helping the team to organize and prioritize goals that would be incorporated into the IEP. From the ICF document, the interprofessional educational and rehabilitation team (i.e., School Principal, Director of Therapy, each student’s special educator, physical therapist, occupational therapist, speech and language pathologist) could more easily prioritize and initially develop goals using GAS methodology. Goals for each student were developed across the categories of physical activity/adaptive physical education, gross and fine motor, language, feeding, cognition, self-care, academics, social and emotional, and transition. Goals were also written across the ICF framework. Once school team members reached a consensus on goals, goals were sent to families for review, revisions, and/or approval prior to the formal IEP meeting. Because of the high level of IEP pre-planning, oftentimes few, if any, changes were made to the goals areas, GASs, or actual IEP document at the final and formal IEP meeting. In cases where GASs were achieved during the school year, the team reconvened to create more advanced goals based on children’s annual assessments. Parents again edited or approved the more advanced goals. Appendix C contains examples of a student’s goal attainment scalings.

### 4.4. Step 4: Goal Measurement across the Year

Each child’s GASs were measured weekly by educational and/or rehabilitation professionals using a specifically created data sheet that was modified from the Murdoch Center Data Sheet from the Murdoch Program Library [39]. See Appendix D for an example. Formal reporting to parents concerning GASs on the IEP data occurred every 9 weeks. If progress reporting indicated the mastery of a GAS with a score of 0 or above, then a more advanced GAS was established. The expected duration of GASs was 1 year, as it aligned with the annual development of IEPs. At the end of the year, raw scores were summed for each student’s set of IEP/GAS goals. This summed raw score is similar to an overall grade point average for a regular education student taking courses in a variety of subjects (e.g., math, English, social studies). The summed raw scores were then translated into scaled and T-scores for each child. Collectively scoring GASs for all children in each classroom and the entire school allowed for T-score calculations for each classroom and the entire school to be generated. These T-scores were reflective of overall classroom and school effectiveness, which were used to inform improvements [11].

## 5. Results

For the 2010–2011 school year, 204 IEP GASs were created for 31 children with disabilities. The average number of goals per student was 6.58, ranging from 5 to 7 goals. Raw scores were converted to T-scores using tables provided in Kiresuk et al.’s book [11] (pp. 274–278). Student, classroom, and overall school scores are displayed in Table 2 and Figure 1. The scores in Table 2 reflect the T-score for each individual child, classroom, and school, having been adjusted for the number of goals a student had during that school year, as students with the same scaled score may have differing T-scores based on their individual number of goals. Based on summed T-scores, seven students demonstrated T-scores of 50 (50 to 71.11), indicating the meeting or exceeding of expectations. Twenty children demonstrated improvement beyond baseline, but below the expected values projected. Four CMD demonstrated no improvement from baseline. One classroom met expectations with a T-score of 50.4. A second classroom was close to meeting expectations with a T-score of 45.1. The remaining classrooms demonstrated improvement beyond baseline, approximating an increase of 1 standard deviation. Reflective of all student goals, the school effectiveness T-score was 35.74.

For the 2011–2012 school year, 192 IEP GASs were created for 30 children with disabilities. The average number of goals per student was 6.19, ranging from 5 to 7 goals. Student, classroom, and overall school scores are displayed in Table 2 and Figure 1. Based on summed T-scores, two students demonstrated T-scores of 50 (50 to 71.11), indicating the meeting or exceeding of expectations. Twenty-four children demonstrated improvement beyond baseline below the expected values projected. Five children demonstrated no improvement. Reflective of all student goals, the school effectiveness T-score decreased between school years to 29.45 with only one classroom improving from the 2010–2011 school year to the 2011–2012 school year.

Because the goals of each child may be unique and the number of goals across children can vary, a statistical analysis of GAS scores is challenging and thus there was no statistical test performed to determine whether a significant difference occurred between school years. In this regard, Figure 2 displays the range of scores for GAS and the traditional assessments CPCHILD, FPS-R, DASH, and PEDI for years 2010–2011 and 2011–2012. A noticeable visual difference occurs both in the range and quantiles of scores within the GAS between the school years, as the scores decreased between years. Of the traditional assessments, only the FPS-R assessment depicts a difference in the scores between the two years; however, the scale to which these ranges of scores differ is far less than that of the GAS scores. The CPCHILD, DASH, and PEDI scales display very little difference between the range of scores from one year to the next.

## 6. Discussion

Goal attainment scaling was able to capture improvement in the majority of CMD. Several or a combination of factors may explain this improvement. First, quality of life discussions with the family provided an opportunity for therapists and teachers to learn what was important from their perspective. Often times in many of these discussions, families highlighted meaningful emerging skills that not only were relevant in the home but also the school and community environments. Second, organizing and prioritizing identified areas of improvement across the ICF allowed family, teachers, and therapists to reach a consensus on goal setting more easily. Third, the use of a battery of assessments, both required and individualized, also helped to recognize and appropriately target emerging skills for goals. Lastly, the weekly formative assessment of goals by scoring the GASs facilitated goal clarification and guided modifications to better target instruction and rehabilitation towards goal success. These explanations for observed GAS improvement are in line with available evidence. In a systematic review of GAS, Haladay and colleagues determined that quantitative, assessment-driven, and patient-initiated goals inclusive of a family-focused approach demonstrated a greater responsiveness than those that were qualitative and provider-initiated [15]. The use of the ICF framework has also documented improved rehabilitation goal setting [40], while goal-directed rehabilitation with regular reassessment has been found to be an effective strategy for achieving goals [41,42].

As stated in the Introduction, measuring change in CMD using traditional educational and rehabilitation measurement tools is challenging, not only because CMD demonstrate varying multi-system impairments and corresponding activity limitations but also because the tools are often unable to capture subtle change that is common in CMD [7,8,9]. Based on our findings, it appears that GAS may provide an opportunity to overcome measurement challenges in CMD. On the other hand, when these measurement tools are able to capture change, it is important to consider that in the context of the home, community, and school, these changes may not be viewed as meaningful by the family and school team. It has previously been noted that GAS provides a greater responsiveness than other well-recognized measurement tools [15]. In this project’s case, although significant improvements were noted for the DASH sensory motor and activities of daily living scales and PEDI Selfcare Caregiver Assistance section, these improvements provide little insight as to each CMD’s individual improvement within the actual IEP as well as meaningful change in the home, school, and community settings.

Some of the children demonstrated no improvement in GAS across the IEP. This is especially true for the 2011–2012 school year when students’ annual IEP was completed on or near a student’s birthday rather than at the beginning of the school year, as was the case in 2010–2011. As such, at the end of the school year when GAS scores were calculated, students with birthdays near the end of the school year had had little time to demonstrate the mastery of goals beyond the current level of performance. Additionally, CMD experience a great number of medical appointments and medical interventions, such as surgeries, hospitalizations, and medication alterations, which often negatively impact school participation and slow the progression of goal achievement. For example, if a CMD were to undergo a posterior spinal fusion in the middle of the school year, then that student may be unable to attend school for 6 weeks. Substantial remediation may also be necessary to regain the preoperative level of function following the return to school. In this case, the progression of skills as defined by goals may be slow because of this absenteeism and/or remediation. Lastly, evidence suggests that in order for teams to create reliable GASs, approximately 3 years of implementation are recommended [19]. Findings from this project spanned across the first and second years of the school’s opening, so the GASs developed by teams may have been less effective GASs, which may have interfered with documenting improvement in goals for CMD.

Just as traditional educational and rehabilitation measurement tools can provide an indication of a program’s effectiveness, so too can GAS. Classroom and overall school performance T-scores derived from GASs accurately revealed the contextual truth occurring within PACE. One informative finding interpreted from the classroom summative scores was the lower T-score seen in classrooms with earlier career or recently graduated teachers and therapists. In the 2010–2011 year with the opening year of PACE, the intentional decision was made to pair early career or recently graduated teachers and therapists within a classroom to foster team development and growth over time. In reflection, a better strategy may have been to pair senior faculty with junior faculty within a classroom. In spite of this possible misstep, information derived from classrooms with lower scores drove professional development. These lower scores allowed PACE leadership to reflect on all student goals within a lower-performing classroom and to clearly identify areas in which additional instructional or rehabilitation education was needed for classroom teams. For example, by examining students’ goals in one of these classrooms, it became quite clear that additional education on cortical visual impairment in CMD was needed. It should be noted that classroom summative scores were never used in a punitive manner as PACE sought to maintain a culture of mutual support with ongoing improvement.

Goal attainment scaling has been effectively implemented in school settings where instruction and rehabilitation co-occur [13,26] and this project adds to this evidence. The previous implementation of GAS in schools has focused on goals specific to the related service areas of physical and occupational therapy within the IEP. The PTCOUNTS [13] study examined physical therapy IEP goals that were related to motor skills and involved children across a variety of diagnostic categories, including CMD. Daly and colleagues [26] also used GAS to successfully measure motor skill improvement following the implementation of an adaptive cycling program for children similar to CMD. This project’s findings specific to CMD expand the current level of evidence by demonstrating that GAS can be used to measure goals across all categories of an IEP by an interprofessional team of educators and therapists in the context of a school setting.

One method of capturing meaning is to measure QOL. With a mission of improving QOL for CMD, at the end of each school year, PACE measured QOL using the CPCHILD. Although no significant changes in CPCHILD scores occurred between 2010–2011 and 2011–2012, scores suggested an elevated QOL for CMD attending PACE. In the CPCHILD manual, the mean CPCHILD value for nonambulatory children considered to be mobility-dependent was 44.4. The CPCHILD scores for the end of the 2010–2011 and 2011–2012 school year were 56 and 59, respectively. With a 5- to 9-point MCID for CPCHILD, the greater than 10-point difference between the reported CPCHILD mean and PACE scores indicates that QOL for PACE students may be elevated. No correlation or causation can be reported as related to this elevation, but one possible explanation may be the capacity of GAS to target and measure change in CMD across meaningful life activities and participation. Knowing that pain can negatively impact QOL, it should also be noted that reported pain severity values for CMD that attended PACE were reduced. The values of 2.92 and 2.15 for the 2010–2011 and 2011–2012 years, respectively, are lower than those reported for similar children [43,44].

Several limitations can be noted within this project. Although derived from a conceptual model, the QOL questionnaire used to facilitate discussion between parents and school team members was not validated. Selection bias was also a limitation as parents chose for their CMD to attend PACE and as such other similar populations of CMD may not display the same results. The addition of a control group that was comparable to the CMD attending PACE would have strengthened this project. The administration of the DASH, CPCHILD, FPS-R, and PEDI at the beginning and end of the school year may have captured a greater degree of change than that observed by only administering them once a year. However, this administrative burden would have been quite time consuming for the school team that would have reduced important instruction and rehabilitation for the CMD. A fifth limitation was the different measurement points between the creation and end of the school year scoring of GAS between the 2010–2011 and 2011–2012 years. The lower GAS scores during the second year may be a reflection of this change in which IEPs were created near each child’s birthday. As a final limitation, the authors acknowledge that these data are older, but children considered multiply disabled or medically complex are a challenging population to study secondary to their heterogeneity and as such there is limited evidence. In the past 10 years, when completing a quick PubMed search for “special education” or “rehabilitation” in conjunction with children with multiple disabilities or children with medical complexity, there have been eight publications. The majority of publications involving this population of children have explored examining hospitalization, care coordination, nutritional support, and pain. in this population.

## 7. Conclusions

Findings from this project provide evidence that GAS is an objective and responsive tool for measuring CMD despite their multi-system impairments and activity limitations. The integration of GAS across IEP categories using an interprofessional team of parents, educators, and rehabilitation therapists can also be successfully executed. The use of GAS in CMD has the potential to maximize participation across the school setting where all children in the United States commonly develop and learn skills as well as find meaning. Teams and schools serving CMD should consider incorporating GAS into practice.

## Figures and Tables

**Figure 1 behavsci-13-00625-f001:**
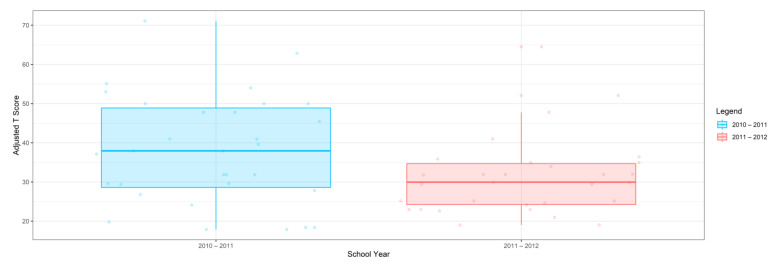
Box plot of adjusted T-scores across school years.

**Figure 2 behavsci-13-00625-f002:**
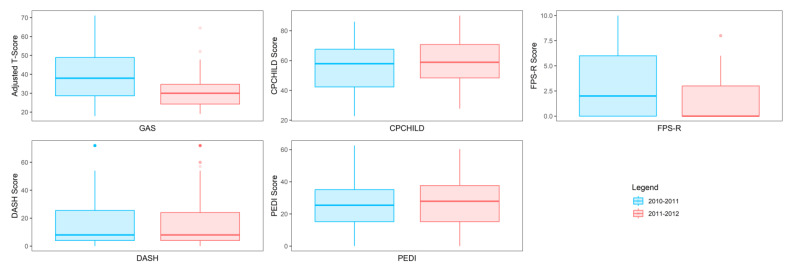
Visualization demonstrating how GAS varies more from year to year relative to traditional assessments (CPCHILD, FPSR, DASH, and PEDI). The vertical axis displays the range of scores for each assessment.

**Table 1 behavsci-13-00625-t001:** Demographics and Characteristics of the Participants (N = 31).

Demographic and Characteristics	# of Children ^1^
DIAGNOSIS	
Cerebral palsy	17
Genetic health conditions	11
Traumatic brain injury	3
CO-OCCURRING CONDITIONS	
Seizures	19
Cortical visual impairment	15
Tracheostomy	1
GENDER	
Male	15
Female	16
RACE and ETHNICITY	
Black/African American	13
White	17
Other	1
Not Hispanic/Latino	31
MOBILITY FUNCTION	
Wheelchair users propelled by caregiver	29
Wheelchair user self-propel	2
FEEDING FUNCTION	
Gastronomy or Jejunostomy tube	13
Gastronomy Tube/Oral	4
Oral	14
SELFCARE FUNCTION	
Maximum assistance	30
Moderate assistance	1
EXPRESSIVE and RECEPTIVE FUNCTION	
Maximum assistance	29
Moderate assistance	2

^1^ One student only attended the 2010-11 school year and not the 2011-12 school year.

**Table 2 behavsci-13-00625-t002:** Goal Attainment Scaling Scores across students for years 2010–2011 and 2011–2012.

	2010–2011 Number of Goals	Scaled Score	T–Score	Classroom T–Score	School T–Score	2011–2012 Number of Goals	Scaled Score	T–Score	Classroom T–Score	School T–Score	Bell Curve of Adjusted Scores Based on Number of Goals
					35.74					29.45	
*CLASSROOM 1*				50.396					32.65		
1	6	0.83	62.91			5	−1.2	31.91			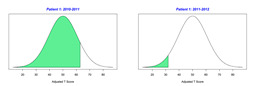
2	6	0.33	44.84			5	−1.2	31.91			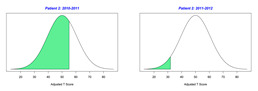
3	5	0.2	53.02			7	−0.86	36.45			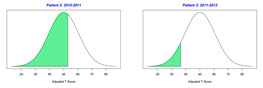
4	7	−2	18.28			5	−1.2	31.91			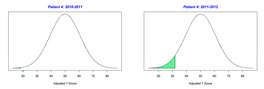
5		0	50			7	−1.14	31.93			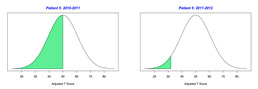
6	7	−0.14	47.74			7	−0.14	47.74			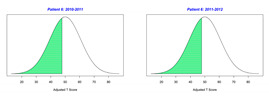
7		0	50			8	−0.88	35.94			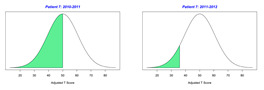
*CLASSROOM 2*				27.356					22.95		
8	7	−1.29	29.67			8	−0.88	35.94			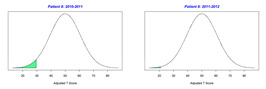
9	6	−0.67	39.67			7	−1.71	22.89			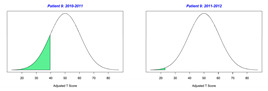
10	8	−2	17.87			5	−1	34.92			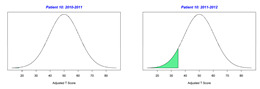
11	6	−1.17	31.93			7	−1.71	22.89			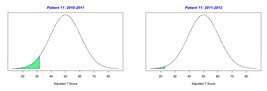
12	8	−1.38	30.12			7	−1.57	25.15			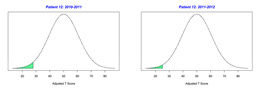
13	7	−1.29	29.67			8	−1.25	29.92			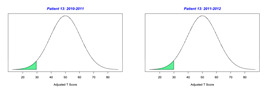
*CLASSROOM 3*				30.413					20.15		
14	5	−0.8	37.94			6	−2	19.02			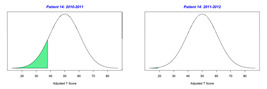
15	6	−1.67	24.18			7	−1.57	25.15			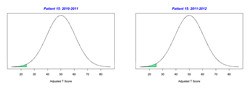
16	5	−0.6	40.95			6	−1.67	24.18			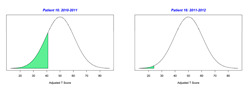
17	7	−2	18.38			6	−2	19.02			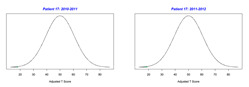
18	7	−0.29	45.48			6	−1.33	29.34			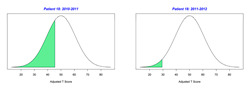
19	8	−1.13	31.93			7	−1.57	25.15			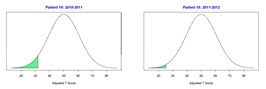
*CLASSROOM 4*				45.058					33.12		
20	7	−0.57	40.96			7	−1.71	22.89			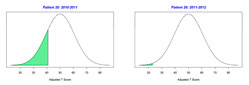
21	6	−1.33	29.34			6	−1.33	29.34			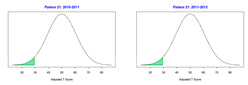
22	8	0.25	54.02			5	−1	34.92			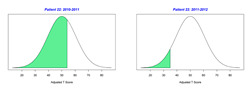
23	5	1.4	71.11			4	1	64.51			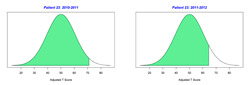
24		0	50				−2	30			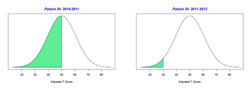
25	8	−0.75	37.95			8	0.13	52.01			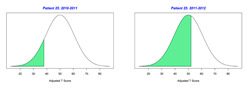
*CLASSROOM 5*				27.513					40.14		
26	6	−1.5	26.76			3	−1.33	31.74			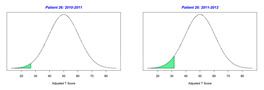
27	8	−2	17.87			8	−1	33.94			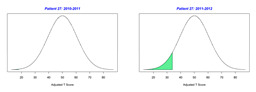
28	6	−1.17	31.93			7	−0.57	40.96			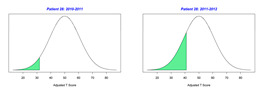
29	7	−0.14	47.74			3	−2	22.62			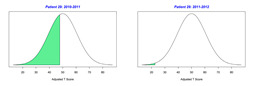
30	8	−1.88	19.88			4	−1.75	24.61			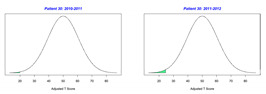
31	6	−0.83	37.09								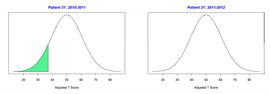

## Data Availability

The authors may contact the corresponding author for data sharing.

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
