# Peer review of "Use of Goal Attainment Scaling to Measure Educational and Rehabilitation Improvements in Children with Multiple Disabilities"

_behavsci, 2023, doi:10.3390/bs13080625_

Round 1

Reviewer 1 Report

Thank you for engaging in this important work. The paper is well-written. Major comments include the need for more detail on the methods, and a question about how the data were analyzed. Also, it seems that there are a few tables missing, along with appendix C.

Total GAS scores were analyzed. Did you folks divide the goals into different categories, ie: QOL, pain and analyze the scores in this manner? It seems like this would be a better comparison as it would better allow you to compare improvements in ADL as measured with GAS with improvements in ADL as measured with DASH’s subdomain score.

Ln 258: Were there a min/max/target number of goals that were set? How were the goals determined / prioritized from the copious amount of information attained from the interview and assessment?  Who determined the goal? How many people were involved for consensus? Did these individuals work with these children? For how long? Was there enough observation / interaction with the child to confidently assign priority to goals? How long did this interview/assessment/creation of goals take for each child?

For the compared assessments that GAS was compared to, were scores of DASH, CPCHILD summed or averaged? For these surveys, were subdomain scores analyzed? Since the GAS sums the scores pertinent to the goals, it is directly measuring the goals, increasing the likelihood of detecting differences. I’m wondering if analyzing subdomains could provide more information vs. using total scores for the entire assessment.

Did the study analyze data from the summer camp? Or year-round school? What was the length of the schooling, ie: 9 month? (ln 118 adds a bit of confusion).

Is there a reference that can be used for ln 121-122, ie: has previous work been done that used this methodology of incorporating 180-min of physically activity? Is it possible to reference this, so the reader can understand the scope of the intervention? Was physical activity the “main intervention” that was used to induce improvement? Or were there other aspects of the intervention? I understand this isn’t the focus of the paper, but it would good what the intervention is so that the reader can estimate the size of expected change/improvement. What time of activity was implemented?

It would be good to see results of DASH, CPCHILD, etc. Also, could you provide examples of the different goals that were set?

Minor comments

Pg 1, Ln 15, “Goal attainment scaling…” this sentence implies children improved, and not their educational practices.

Ln 34: Consider using “typical children” as children considered typically developing is a bit confusing. 

Ln 55:  I’m not familiar with IDEA, so I’m a bit surprised it includes individuals up to 21 years of age, as typically individuals >18 years are considered adults. Does assume that the child may take longer to achieve benchmarks?

Ln 93: first mention of CP? Cerebral palsy? Ln 101 – here it is spelled out.

Ln 111 – Even though the study was considered quality improvement and consent with attained, was IRB approval obtained? Or did the board not consider this retrospective research to require approval? An explanation would be helpful.

Ln 114: This sentence implies this school only serves atypical children. Or were there other classrooms that served neurotypical/typical children?

Ln 125, add space before “1 speech”

Length of time spent at the school?

Ln 162: T scores are, replace “is”

Ln 187, 189 – there are extra spaces

Ln 217 – first use of CVI? (need to define here?), also there’s an unnecessary coma after “CMD”

Ln 232: add “is” before “used”

Ln 239: missing period and a reference here would be appropriate.

Ln 264 – I  may have missed it, but where is Appendix C?

Ln 267 – if GAS was achieved, how was a more advanced GAS determined?

Ln 268 – Were the scores assessed at the assessment time point (every 9 wks) or over a one-year period? If it’s the latter, wouldn’t the summed raw scores not accurately represent changes (as new goals could be set which could underestimate the child’s score)?

Table 2 -font is very tiny and I can’t read the axes titles on the graph. Is column one subject number?

Ln 293: display instead of “displays”

Ln 293, regarding this paragraph, as these assessments measure certain areas, did you compare changes with GAS of similar areas with these assessments? Was it due to lack of sensitivity of the DASH, CPCHIID, FPS-R, or that GAS allowed for more specific goals to be set? Ln 321 – there’s mention of this here, but my question is: If a child’s goal is related to self-care, would the PEDI be able to measure changes? Is it due to the lack of sensitivity of the PEDI (i.e., scoring), or that the PEDI does not directly assess improvement in the area which has improved? Ln 333 suggests it’s both reasons (lack of sensitivity and the surveys are not specific to the goal). I ask because based upon your answer, it would suggest to omit the use of DASH, PEDI, CPCHILD altogether in this population. Would there be a circumstance where DASH, PEDI, CPCHILD could be used? For instance, the methodology undertaken in this study was quite intensive, and for schools lacking resources, is there a workaround? What could you recommend in this  regard?

Tables 3-5: I can’t seem to locate them in the paper.

Ln 557- what is “Les?”’ Also in this appendix, spacing is off for some bullet points.

For subjects who displayed improvement the first year, but did not show as much improvement the next year, what do you think the reason is? Ie: diminishing returns, lofty goals?

None - English language is good. 

Reviewer 2 Report

Thank you for the opportunity to review this manuscript. Here are some of my comments for your consideration.

Table 1: The last column is the number of students. I suggest author(s) add table header to the top of the table, so it’s clear to readers what content of each column is. Please remove the average age from the top of the table to the table note because it’s not the number of students presented in other rows.

Line 170: The section 4 procedure describes 4.1. 4.2-4.5 were explaining how to do 4.1.  I would suggest removing subheading heading 4.1

Lines 174-175: author(s) stated “three steps…..”  Step 4 was listed as 4.5.  Please verify how many steps?

Line 264: no appendix c included in the manuscript.

Table 2: not sure how the T-scores were calculated.  I used the all 31 scale scores to calculate their z-scores and convert it to T-scores, my t-scores are different from what author(s) provided. I also tried to use all 62 scale scores to calculate z-scores and T-scores, and my scores are different from the scores in Tables 1. Please check and provide the formula from lines 161-162 Goal Attainment Scaling: Applications, Theory, and Measurement (p.274) on how the scores were calculated. How was the school’s T-score calculated? I tried the app and did not find out how it was calculated. Please elaborate. Also, lines 161-162 stating convert the scores to T-score, but never mentioned about scale score in Table 2. Author(s) need to describe how the scale score was calculated and the relationship among raw score, scale score, and T-score.

In the procedure subsection, I suggest author(s) provide brief information on how the data were collected and analyzed, the length of the measure, the range of scores, and so on. Because there is no analysis subsection provided in the method section, this information should be added here. For example, how the family interview results were analyzed. Please check all the measures listed in procedure subsection and add the information.

Results

I found it’s hard to follow between the first and second paragraph.  It stated results by number of students and classroom in the first paragraph, but only student-level results were mentioned. Also, only 18 students were mentioned in the first paragraph. What are the outcomes for the other three students?

Line 293: I suggest stating the content of each table separately.  It’s hard to follow and find what table includes which measures. Also, there are two “ands” in this sentence.

Line 196: Please add the abbreviation PEDI after “Pediatric Evaluation of Disability Inventory,” so it’s easier for reader to follow the abbreviation term in the result section (e.g., line 294)

Line 300: Please add table title to Table 2.

Line 310: Please add table title to Table 3. The scales listed in Table 3 do not match the five scales listed in lines 181-182. Please add the possible score range of each scale in the paragraph of the description of  the DASH 2, so reader can understand what the numbers listed in Table 3 meant.

Tables: The number of decimal places should keep consistent across all tables.

Line 294: author(s) stated “Of the 5 Dash sections…..” However, in line 181, author(s) stated “… across five scales.”  Please use the same terms to describe these 5 scales/ sections throughout the manuscript.

Lines 312: The results of Tables 4 (lines298-299) should be described before Table 5(lines 296-298) to make the order consistent as the order listed in Lines 293- 294, so it’s easier for reader to follow.

Line 299: author(s) stated “no significant changes …..” what kind of the analyses were conducted? Please add an analysis subsection.  If it fits better in the procedure subsection, you may consider adding texts in there.

I would suggest author(s) adding another paragraph in the result section (not sure how it should look like since no idea how the measures were analyzed. If they were quantitative data, regression analysis will be recommended) to make connections between the GAS and the other measures, in addition to describing them separately. Please add texts to the discussion section to address any implications to the future study from the perspective of this study.

In the procedure section, several measures/tools were administrated to develop GAS. This would be very helpful for the practice if author(s) can provide more information or examples in this process. For example, provide GAS for the example in Appendix B. 

One final note, isn’t the data too old? Which was more than a decade ago.  Can they still be useful? Or none has done this before in this field? Please address this in the discussion.

Reviewer 3 Report

GENERAL COMMENTS:

I would like to congratulate the authors of the manuscript for their work. The use of the GAS-type methodology to evaluate the effectiveness of an intervention is something to be highlighted. However, there are certain aspects that i believe should be modified to facilitate the understanding of the article.

INTRODUCTION

The introductory section is very complete and tailored to the reader's needs. I do not think it is necessary to modify or include a section in this section.

METHODS

To clarify the wording of the GAS objectives, I believe it would be appropriate to include a table in the body of the manuscript in which an objective is exemplified, so that it can be seen how progress is quantified and graded at levels from -2 to +2.

DISCUSSION

The discussion section adequately justifies the findings obtained in the study, so the information provided is excellent.

REFERENCES

References in the text should be made following the methodology indicated by the journal, i.e., indicating the numbers in square brackets, and not placing them in a superindex.

Round 2

Reviewer 1 Report

Thank you for providing edits and for continuing to invest your time into this important work. Thank you for responding to my initial comments, and providing access to the figure/tables/appendices which has provided clarity. Please see minor comments below.

Ln 52: consider adding “,” after “setting”

Ln 59, assessments, plural?

Ln 94, 96, 97, 179 – did you mean to add an s after GAS?

Ln 105, consider using past tense, aim of the project was to describe…

Please check spacing throughout, ie: ln 117.

Ln 167 , consider adding “the” before GOALed app.

Ln 197 – is interaction intended to be hyphenated?

Ln 253, consider adding “a” before “script”

Ln 358, consider adding a d, “patient-initiated”

Ln585: lower is missing an r

Minor edits are needed, as well as a final read though to ensure proper punctuation, spacing.
